# A building-scale modeling framework for urban net-zero transitions in Nanjing

Yuxin Chen[1], Zhenyu Wang[1], Quan Wen[1], Jing Meng[2], Jingwen Huo[2], Shuping Li[1], Li Zhou[3], Peipei Chen[4], Diling Liang[2], Jun Bi[5] & Dabo Guan[1,2] ✉

Carbon reduction during the operational phase of buildings is a critical component in achieving global carbon neutrality objectives. Current emission estimation methods often overlook building-level heterogeneity, limiting precise retrofit strategies. Here, we develop a building-based emissions accounting framework incorporating building typology, function, and geometry, augmented by facility-level power plant data. We propose tailored operational-phase mitigation technologies, analyzing 2020-2050 pathways through baseline, regulatory, and blueprint scenarios. Demand-side strategies target energy behavior modification (e.g., efficient lighting), while supply-side interventions prioritize coal-to-biomass conversion and fossil plant retirement. Applied to Nanjing (534,000 buildings across 101 streets), results show commercial buildings exhibit 3.9 times higher carbon intensity than residential units. End-use efficiency upgrades (HVAC, lighting, appliances) prove most effective for commercial sectors, whereas supply-side gains derive primarily from accelerated coal plant phaseout before 2045 and renewable integration (solar/wind/nuclear). This approach provides actionable building-specific decarbonization pathways, offering policymakers science-backed strategies for urban energy transitions.

Climate change represents a critical and pressing challenge, increasingly recognized by the global community[1–4]. In 2022, carbon dioxide emissions (hereinafter referred to as "$CO_2$ emissions") from building operations and construction activities reached a historic peak of 10 Gt, representing 37% of global energy-related $CO_2$ emissions[5]. Notably, the building sector within the construction industry remains a substantial source of emissions, offering considerable potential for reductions[6]. In China, the construction sector is particularly pivotal, given its high energy consumption and substantial share of $CO_2$ emissions[7]. Specifically, China's building operational emissions reached 2.3 Gt $CO_2$ in 2021, constituting 21.6% of national energy-related emissions and 56.6% of total whole-lifecycle building sector emissions (4.07 Gt $CO_2$)[8]. Consequently, decarbonizing the operational phase of buildings is essential for achieving global carbon neutrality targets[9,10].

Decarbonizing end-use sectors is fundamental to achieving carbon neutrality in buildings[11–13]. This process necessitates the incorporation of micro-level information such as building type, function, footprint area, and floor height for accurate identification and strategic planning. However, much of the existing research adopts a "top-down" approach to chart future decarbonization pathways for buildings[14,15], employing methods like input-output models[16–19] and computable general equilibrium models[20,21]. While these "top-down" approaches are useful for studying the relationship between national economies and energy use, forecasting $CO_2$ emission trajectories, and informing policy development, they are inherently macro-level and lack the granularity needed to delineate specific pathways for achieving carbon neutrality at the building scale. On the other hand, "bottom-up" simulation methods such as the MARKAL/TIMES model[22], MESSAGE model[23], EFOM-ENV model, and LEAP model[24–26] have been employed to forecast future energy demand and $CO_2$ emissions across various sectors. Despite their detailed focus, these studies often fall short of integrating energy

[1]Department of Earth System Science, Tsinghua University, Beijing, China. [2]The Bartlett School of Sustainable Construction, University College London, London, UK. [3]China Renewable Energy Engineering Institute, Beijing, China. [4]Cambridge Judge Business School, University of Cambridge, Cambridge, UK. [5]School of the Environment, Nanjing University, Nanjing, Jiangsu, China. ✉e-mail: guandabo@tsinghua.edu.cn

demand-side and supply-side factors to comprehensively analyze energy consumption and the associated $CO_2$ emissions[27](SI Table 1). As critically identified by Keirstead et al.[28], while most building energy system studies (~70%) incorporate endogenous demand-side parameters, supply-side factors remain systematically under-represented. When occasionally included, they are typically reduced to static carbon intensity factors for rudimentary energy-to-emission conversions[28]. Although Zhou et al.[29] pioneered an integrated framework incorporating both demand-side and supply-side mitigation policies, their analysis remained constrained at the national level in China, without resolving critical spatial scales down to urban systems −let alone neighborhood or individual building levels that are essential for implementable decarbonization strategies. This gap underscores the need for a more holistic framework that bridges demand-side dynamics with supply-side considerations to achieve precise and actionable decarbonization strategies for buildings. In this paper, we develop a building-based emissions accounting approach at the city-scale that considers building type, function, footprint area, and floor height of individual buildings, and we integrate point source data from facility-level power plants to characterize the structure of the energy supply sector. In this way, the $CO_2$ emissions of Scope 1 and Scope 2 in the operational phase of the building are calculated.

To effectively target emission reductions during the operational phase of buildings, precise $CO_2$ emissions calculations at the building scale are crucial[30]. However, current research on building decarbonization predominantly focuses on global regions[31–33], national levels[29,34], their inherent reliance on aggregated data limits actionable insights for city-scale interventions. While some studies have examined emissions at the individual building scale[35], these often emphasize the impact of specific technologies rather than exploring the integration and synergy of multiple technologies[36,37]. Besides, although the analysis of individual buildings can achieve precise accounting results, it is difficult to integrate enough building cases, so it is impossible to provide planning suggestions at the intermediate level of administrative divisions. There is a significant gap in research that integrates a comprehensive suite of emission reduction technologies to assess energy consumption and $CO_2$ emissions with high precision in building-based during their operational phase. To address this gap, we propose a suite of emission reduction technologies tailored for the operational phase of buildings, constructing various scenarios to analyze the emission reduction pathways. Our building-level approach advances this field by: (i) establishing a bottom-up accounting framework for operational energy consumption and $CO_2$ emissions at the individual building level, thereby bridging the critical "last-mile gap" that persists between macro-scale urban decarbonization targets and micro-scale building operational practices; (ii) enabling targeted identification of critical emission sources through POI-based building classification coupled with physical attributes (e.g., height, footprint area); and (iii) providing integrated demand−supply mitigation pathways that operationalize policy mandates into implementable solutions, addressing the critical limitation in some studies which focused solely on single-dimension approaches (e.g., [Sarica et al.]'s HVAC efficiency[38]; [Kannan and Strachan]'s renewable energy supply[39]; [Xiao et al.]'s Disruptive decarbonization technology: Carbon Capture, Utilization and Storage[40]) without systemic coupling.

In this work, we develop a comprehensive model−the CEADs-Building Model−for building-based emission accounting and the analysis of emission reduction pathways. This model integrates several key components: an energy demand module, an energy conversion module, an environmental impact module, a socioeconomic module, and an emission reduction module. The energy demand module is designed to account for different building types, incorporating specific terminal energy consumption parameters to simulate both current and future energy demands during operational phases. The

energy conversion module integrates facility-level power plant data to characterize the structure of the energy supply sector. By linking demand-side and supply-side, the model calculates emissions from both Scope 1 (direct) and Scope 2 (indirect) sources. The environmental impact module applies IPCC methodologies to calculate $CO_2$ emissions using emission factors for various energy types. The socioeconomic module projects future trends in building areas by combining macroeconomic data(such as GDP and population) with per capita residential and public floor area metrics. The emission reduction module classifies technologies from both demand-side and supply-side perspectives, simulating future emission trajectories from building energy use under three scenarios: baseline, regulatory, and blueprint, each reflecting the impact of different technological configurations. On the demand side, the pathways emphasize two primary categories of emission reduction technologies: reducing service demand and enhancing energy efficiency across 15 types of end-use energy equipment. On the supply side, strategies include improving power generation efficiency, optimizing the energy structure, and enhancing load flexibility (Supplementary Fig. 1). Detailed methodologies and datasets used in the development of the CEADs-Building Model are provided in the methodology section.

## Results

### Current state of $CO_2$ emissions from the demand-side during the operational phase of buildings in Nanjing

In 2019, the total $CO_2$ emissions from the operational phase of buildings in Nanjing amounted to 19,015.37 thousand metric tons of $CO_2$ equivalent (hereinafter referred to as "kt $CO_2$e"). To precisely identify decarbonization priorities, a functional and sub-district disaggregation of emissions is essential, as building types (e.g., residential, commercial) and micro-zones exhibit fundamentally distinct emission profiles. Spatially, these emissions exhibited a pattern of uniform distribution within the city center, while displaying a more dispersed, multifocal distribution in suburban areas (Fig. 1a). In terms of emission intensity, the city's average $CO_2$ emission intensity per unit floor area was 6.48 kg $CO_2/m^2$. By conducting $CO_2$ emission accounting at the building scale and incorporating the buildings' specific attributes and functions into our analytical framework, we identified significant variations in $CO_2$ emission intensity among buildings with different functions during their operational phase. The subsequent analysis focuses on several representative streets to further explore these disparities.

The Nanjing Economic and Technological Development Zone (NETD, northeast of Nanjing)−contributing 32.7% of the city's industrial GDP and housing 18% of its manufacturing facilities−exhibits substantial overall emissions, coupled with high emission intensity (Fig. 1b). This can be attributed to the area's predominantly commercial and industrial building types and land uses, leading us to classify it as a "Industrial base street" for decarbonization studies. In 2019, $CO_2$ emissions during the operational phase of buildings in this zone reached 632.54 kt $CO_2$e, with an emission intensity of 10.63 kg $CO_2/m^2$, which is 1.64 times higher than the city's overall carbon intensity. This elevated emission level is largely due to the concentration of industrial enterprises in the NETD, including sectors such as optoelectronics, high-end equipment manufacturing, biopharmaceuticals, modern logistics, and technological innovation. Among the 10,475 individual buildings in the zone, 65.77% are commercial buildings, characterized by dense factories and large building areas. These factors contribute to significant energy consumption, as calculated using a "bottom-up" approach focusing on end-use energy sectors and equipment, which in turn leads to higher $CO_2$ emissions. For instance, a 14-story building within Nanjing Baori Steel Wire Products Co., Ltd., located in this zone, generated 978.23 t $CO_2$e in 2019.

Figure 1c illustrates that Moling Street(south of Nanjing) exhibits relatively high overall emissions, with emission intensity exceeding

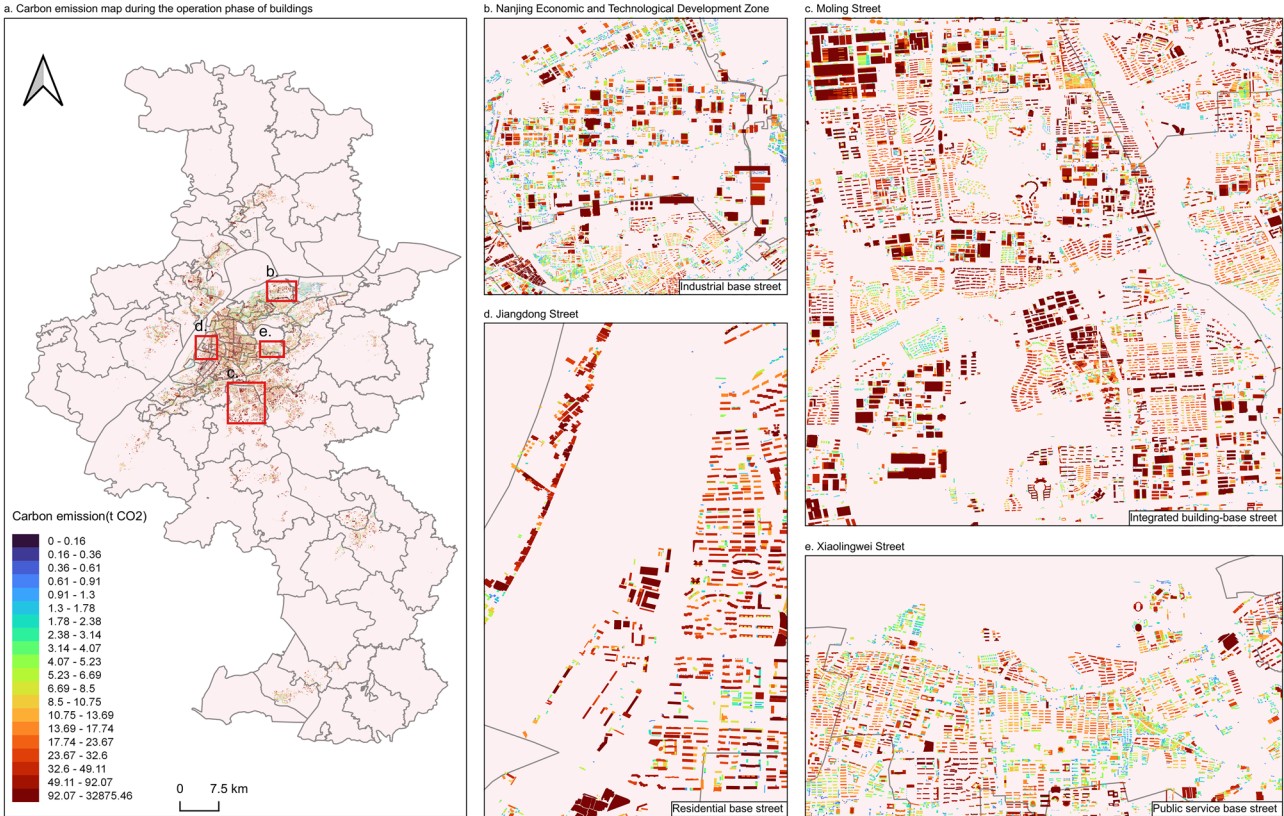

**Fig. 1 | Current state of CO₂ emissions from the demand-side during the operational phase of buildings (2019). a** illustrates the CO₂ emissions during the operational phase of buildings across Nanjing City, based on comprehensive building data. The area map in the upper left corner highlights the geographical location of Nanjing. **b–e** Depict the CO₂ emissions associated with energy consumption during the operational phase of buildings in specific areas: Nanjing

Economic and Technological Development Zone, Moling Street, Jiangdong Street, and Xiaolingwei Street, respectively. The base map was created in QGIS (v. 3.28.11-Firenze; https://www.qgis.org) using data from the Standard Map Service of the Ministry of Natural Resources of China (Review Map No.: GS(2019)1822; http://bzdt.ch.mnr.gov.cn/). Copyright of the base geographic data belongs to the Ministry of Natural Resources.

the city average but remaining lower than that of the NETD. This pattern likely results from Moling Street's extensive coverage and diverse building types, so we classify it as an "Integrated building-base street." Specifically, CO₂ emissions during the operational phase of buildings in Moling Street amounted to 1809.95 kt CO₂e, with an emission intensity of 6.99 kg CO₂/m². While the carbon intensity in Moling Street surpasses the city's overall level, it is still below that of the NETD. This difference is primarily due to Moling Street's larger jurisdiction, encompassing nearly four times the number of buildings found in the NETD. The street includes a mix of residential, commercial, and public buildings, in contrast to the predominantly commercial structures in the NETD. The deep red areas in Fig. 1c represent industrial buildings, with Chang'an Mazda Automobile Co., Ltd. Located within Moling Street, Chang'an Mazda comprises 233 buildings and was responsible for 89.09 kt CO₂e in 2019.

The overall emissions from Jiangdong Street(west of Nanjing) are relatively low, with a correspondingly low emission intensity (Fig. 1d). Although Jiangdong Street, situated in the City Center, has the highest emissions within the Gulou District, its CO₂ emissions during the operational phase of buildings amount to 114.68 kt CO₂e. This figure is modest when compared to the emissions from NETD and Moling Street. The emission intensity in this area is 5.42 kg CO₂/m², which is below the city's average carbon intensity. The street's proximity to the Jiangsu Provincial Government and the fact that 75% of the buildings are residential, which have a significantly lower carbon intensity than commercial buildings, contribute to its relatively low overall emissions and intensity. In contrast, Xiaolingwei(east of Nanjing) Street has the smallest total emissions and the lowest emission intensity among the

four selected cases (Fig. 1e). The CO₂ emissions during the operational phase of buildings in Xiaolingwei Street total 151.48 kt CO₂e, with an emission intensity of 4.50 kg CO₂/m². This street predominantly serves as a hub for scientific research and education, housing institutions such as the Jiangsu Academy of Agricultural Sciences, Nanjing University of Science and Technology, and Nanjing Agricultural University. Additionally, Xiaolingwei Street—a major academic hub hosting 12 research institutes—exhibits distinct energy patterns: its lower average building height (8.58 vs. 12.36 floors in Jiangdong Street) reduces HVAC loads. Consequently, despite having more than twice the number of buildings as Jiangdong Street, Xiaolingwei Street's total CO₂ emissions are only 1.32 times higher. Based on these characteristics, we have classified Jiangdong Street as a "Residential base street" and Xiaolingwei Street as a "Public service base street".

## The CO₂ emission sources of different functional streets vary greatly

In the NETD, commercial buildings are the dominant source of emissions, contributing 90.10% of the total, which aligns with the street's predominant land use for commercial and service purposes. Specifically, the primary sources of emissions within these buildings are commercial HVAC systems, commercial power sectors, and commercial lighting, which generated 222.08 (35.11% of the street's total emission), 110.20 (17.42%), and 87.77 (13.88%) kt CO₂e, respectively, in 2019. For Jiangdong Street, where 75% of the buildings are residential, the overall carbon intensity of the residential sector (3.36 kg CO₂/m²) is lower than that of public buildings (10.52 kg CO₂/m²). However, due to the large number of residential buildings, their total CO₂ emissions

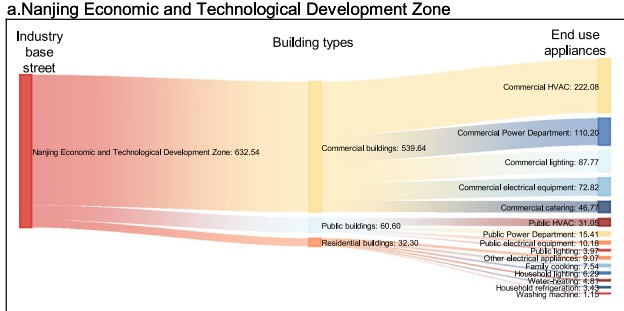

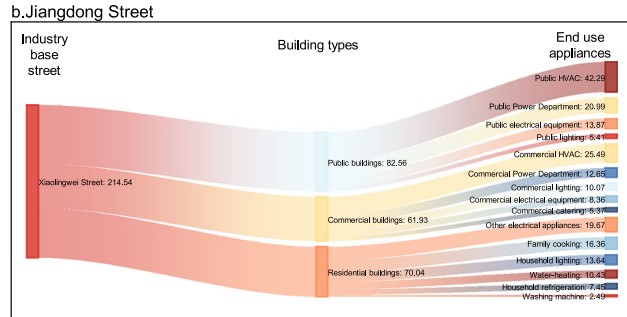

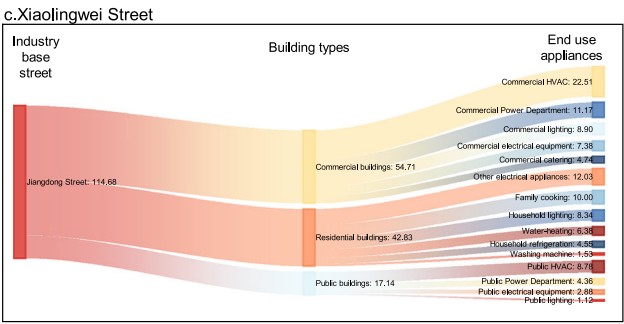

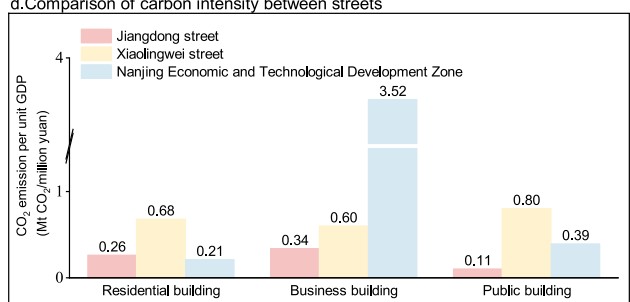

**Fig. 2 | Comparison of CO2 emission magnitudes and intensity by building types and end-use appliances. a–c** Depict the CO2 emissions associated with energy use during the operational phase of buildings in NETD, Jiangdong Street, and Xiaolingwei Street in 2019, categorized by "street-building type-terminal energy department." **d** Presents a bar chart comparing the CO₂ emissions per unit GDP across the three streets.

surpass those of public buildings, making residential buildings the second largest source of emissions by building type. Consequently, the primary focus for emission reduction in Jiangdong Street should be on residential buildings. In contrast, public buildings are the main contributors to emissions in Xiaolingwei Street, with 82.56 kt CO₂e emitted in 2019. Within this sector, public HVAC systems alone accounted for 51.22% of the total emissions from public buildings, equating to 42.29 kt CO₂e. This high level of emissions is largely due to the concentration of scientific research and educational institutions in Xiaolingwei Street, including universities, primary and secondary schools, research institutes, and their associated facilities.

The comparison of carbon intensity (CO₂ emissions per unit GDP) of different building types in the three aforementioned streets mentioned above is shown in Fig. 2d. We calculated the CO₂ carbon intensity using the GDP seat economy index of the administrative district where each street is located. Among residential buildings, the carbon intensity in the NETD are the lowest, at just 0.21 Mt CO₂/million yuan. This can be attributed to the development zone's primary focus on industrial and commercial activities, with relatively limited residential functions. The scarcity of residential buildings results in lower overall CO₂ emissions and carbon intensity for this category. In Jiangdong Street, the carbon intensity for residential buildings is slightly higher than in the NETD but remains relatively low. Although Jiangdong Street is primarily a residential and consumer-oriented area, its proximity to the Jiangsu Provincial Government means the surrounding regions are dominated by high-value-added consumer service industries, which contribute to a high GDP. However, the carbon intensity of terminal energy equipment used in residential buildings is comparatively low. These combined factors lead to a modest overall carbon intensity for residential buildings in Jiangdong Street.

For commercial buildings, the ranking of carbon intensity across the three streets is as follows: NETD, Xiaolingwei Street, and Jiangdong Street, with nearly a tenfold difference between the highest and lowest values. This disparity is primarily due to NETD's focus on high-value-added manufacturing and trade services, attracting numerous

industrial enterprises in sectors such as biomedicine, optoelectronics, modern logistics, and high-end equipment. These commercial buildings are equipped with high CO₂ emission intensity terminal energy-consuming systems, such as HVAC systems and commercial electrical equipment, resulting in the highest carbon intensity in NETD. This finding aligns with previous research indicating that the electricity consumption of large commercial buildings can be many times greater than that of residential buildings, suggesting that the economic cleanliness of NETD is lower compared to the other two streets. In the case of public buildings, Xiaolingwei Street exhibits the highest carbon intensity, at 0.72 Mt CO₂/million yuan. This outcome is attributable to the fact that most of the land in Xiaolingwei Street is designated for public use, with public buildings having high-energy-intensity terminal equipment, leading to elevated CO₂ emissions. Additionally, the economic development level of the administrative district in which Xiaolingwei Street is located is lower than that of Jiangdong Street ("Residential base street") and NETD ("Industrial base street"). These factors result in significantly higher carbon intensity for public buildings in Xiaolingwei Street compared to the other streets.

## Large coal-fired power plants are the predominant sources of emissions on the supply-side

Figure 3a provides a comprehensive overview of the operational lifespan, installed capacity, fuel type, and power generation technologies of Nanjing's generator units. The figure highlights that the majority of Nanjing's coal-fired power plants predominantly utilize subcritical power generation technology. Notably, coal-fired units with an operational lifespan of 12–20 years account for 73.13% of the total installed capacity, indicating that most of these units are relatively new. A significant expansion of Nanjing's power infrastructure occurred between 2009 and 2010, marked by the addition of a 3380 MW ultra-supercritical power plant, which was driven by China's national economic stimulus plan implemented to counter the global financial crisis, while simultaneously addressing Jiangsu Province's rapid

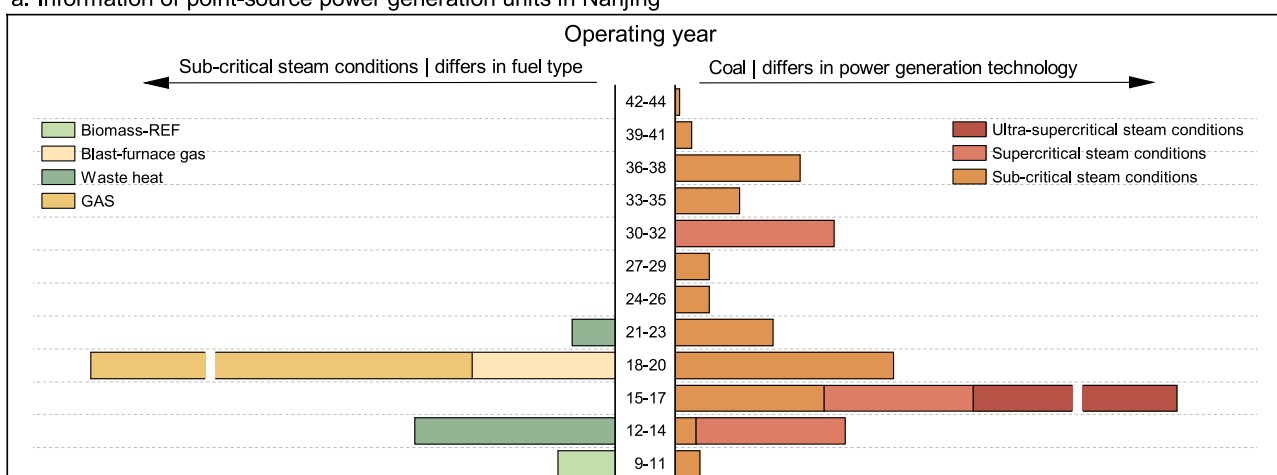

a. Information of point-source power generation units in Nanjing

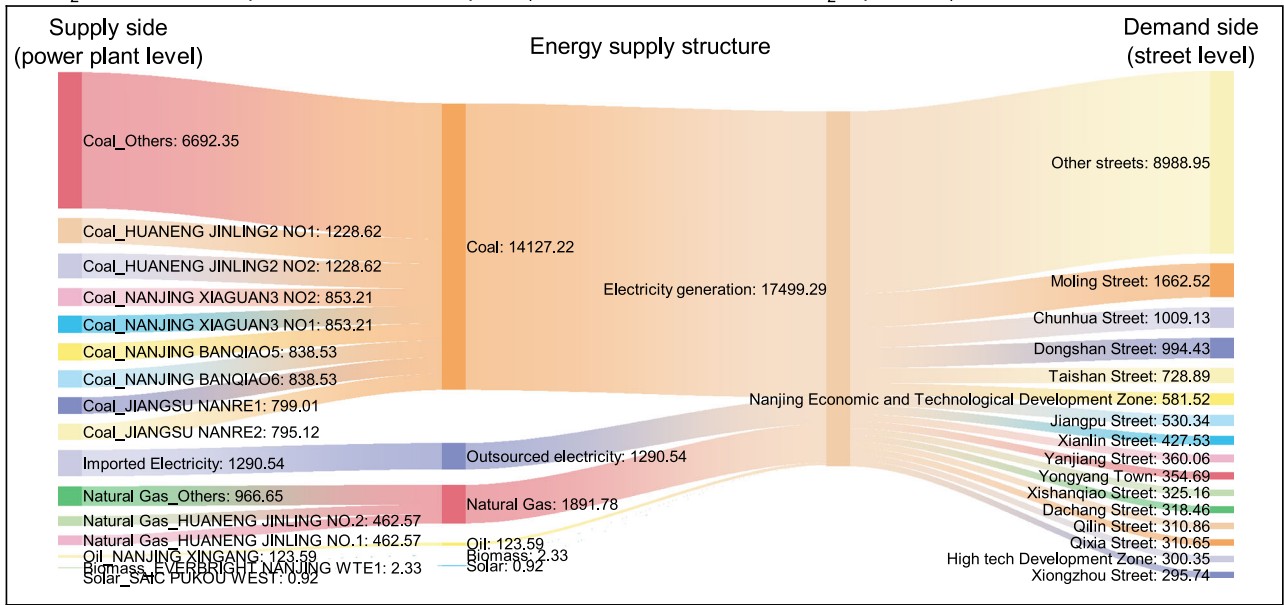

b. CO₂ emissions from production to consumption (Thousand Metric Tonnes CO₂ equivalent)

**Fig. 3 | Current status of supply-side generator units and CO₂ emissions from the "supply–demand" side during the operational phase of buildings in Nanjing (2019).** The *X*-axis of the **3a** represents the installed capacity of the various units, while the *Y*-axis indicates the operational duration of these units, spanning from their commencement of production up to 2024. The left side of (**a**) illustrates all power plants utilizing subcritical power generation technology, with different fuel types of generator sets distinguished by unique colors. On the right side, **a** focuses exclusively on coal-fired power plants, using distinct colors to differentiate between generation technologies, including subcritical, supercritical, and ultra-supercritical. **b** Depicts the coupling of energy supply and demand during the operational phase of buildings in Nanjing in 2019. On the demand side, emissions were calculated for 101 streets in Nanjing, with the top 15 streets with the highest emissions highlighted individually, and the remaining streets are grouped under the "Other streets" category.

industrial growth[41]. This dual national-local context accelerated the adoption of advanced coal technologies, explaining both the project's timing and its technological specifications. Furthermore, between 2005 and 2007, Nanjing introduced a large-scale gas-fired power plant, utilizing both blast-furnace gas and natural gas, with an installed capacity of 880 MW. It is also noteworthy that over the past decade, there has been minimal expansion in coal-fired power plants in Nanjing, alongside a pronounced shift toward waste heat recovery technologies, which accounted for 150 MW of new capacity in 2018 alone. This transformation has been driven primarily by Jiangsu Province's 13th 5-Year Energy Plan (2016–2020), which set a binding target of 15% industrial waste heat utilization[42], and reinforced by Nanjing's

Municipal Low-Carbon Development Ordinance prohibiting new coal-fired power plants within urban core zones[43].

The power sector plays a central role in the supply-side of emissions. In this study, we incorporated 66 operational power generation units in Nanjing into the "bottom-up" model to represent the supply-side. In 2019, the total installed capacity of these local power generation units was 12,626.46 MW, with coal-fired units comprising 79.23% of this capacity. The contributions from oil-fired, solar, and biomass power plants were minimal. Our calculations indicate that in 2019, the overall CO₂ emissions during the operational phase of buildings in Nanjing amounted to 19,015.37 kt CO₂e. Of this total, indirect emissions from the power sector were 17,499.29 kt CO₂e, accounting for

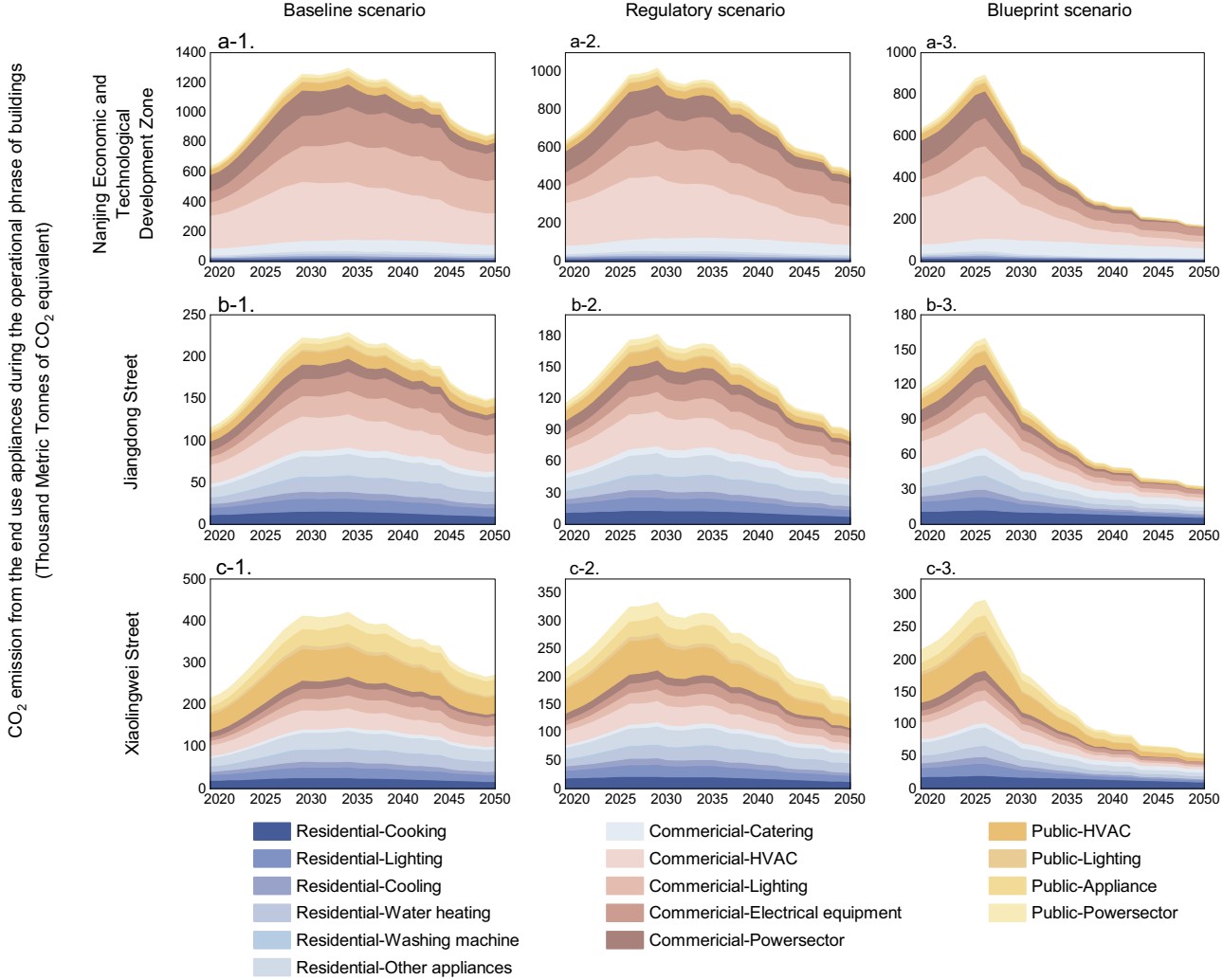

**Fig. 4 | Comparison of emissions across multiple scenarios under CO₂ emission reduction technology pathways during the operational phase of buildings in Nanjing. a-1**, **a-2**, and **a-3** present the CO₂ emissions of end-use energy-consuming sectors during the operational phase of buildings in NETD from 2020 to 2050 under the baseline, regulatory, and blueprint scenarios, respectively. Similarly, **b-1**, **b-2**, and **b-3** depict the corresponding emissions for Jiangdong Street across these scenarios. **c-1**, **c-2**, and **c-3** illustrate the CO₂ emission trends for Xiaolingwei Street under the same set of scenarios.

92.03% of the total (as shown in Fig. 3a). Notably, eight coal-fired units each emitted more than 1000 kt CO₂e annually. These high-emission units are operated by major power companies, including China Resources Power Holdings Company Limited, Datang Jiangsu Power Generation Co., Ltd., and HUANENG Power International Inc.

Figure 3b integrates the energy supply-side with the demand-side, illustrating CO₂ emissions during the building operation phase from both perspectives. The supply-side is represented by power plant facility-level generators, while the demand-side is modeled as various streets within the city. The analysis reveals that on the supply-side, the primary sources of emissions are concentrated in coal-fired power plants, which contribute 80.73% of the total emissions from all fuel-type power plants. Notably, two coal-fired generators operated by HUANENG Power International Inc., with a combined installed capacity of 1030 MW, produced 1228.62 kt CO₂e in that year. Additionally, Nanjing's external electricity purchases also contributed significantly to emissions, amounting to nearly 1290.54 kt CO₂e. On the demand side, the top street contributing to emissions is Moling Street(with emissions of 1809.95 kt CO₂e), while the NETD ranks fifth among all streets. Collectively, the top ten streets account for 39.85% of the city's total emissions, highlighting these areas as critical targets for emission reduction efforts within Nanjing.

## The impact of different emission reduction strategies varies significantly

As shown in Fig. 4a-1, a-2, a-3, using NETD as an example, the cumulative and peak CO₂ emissions generated by energy consumption during the operational phase of buildings differ significantly across scenarios. In the baseline scenario, the energy intensity and performance of public buildings are primarily driven by the level of energy-saving technology associated with China's future socio-economic development. Consequently, CO₂ emissions exhibit an initial increase followed by a decline, peaking at 1294.07 kt CO₂e in 2034, and gradually decreasing to 856.06 thousand metric tons by 2050. Under the regulatory scenario, which emphasizes green and sustainable development, China aggressively promotes energy-efficient retrofitting of public buildings. Additionally, with moderate growth in population and GDP, coupled with an optimized industrial structure, energy consumption intensity and levels decrease relative to the baseline scenario. As a result, CO₂ emissions increase slowly, peaking at 1014.85 kt CO₂e in 2029, before gradually declining. The blueprint scenario represents an optimized approach aimed at achieving "carbon neutrality" by further advancing the development of energy-saving technologies. In this scenario, CO₂ emissions during the operational phase of buildings peak earlier, in 2026, and decline steadily to 52.95 kt CO₂e by 2050.

Figure 4a-2, b-2, c-2 shows that $CO_2$ emission reduction priorities vary across different streets. Specifically, "Industrial base street" need to focus on commercial and service buildings for emission reduction. Taking regulatory scenario as an example, in the NETD, which is mainly focused on manufacturing and trade services, commercial and service buildings are the main type of buildings, and the terminal energy-consuming equipment in these buildings has the characteristics of high energy consumption and high emissions. Key areas for emission reduction include the electrical equipment and HVAC systems in these buildings. For instance, emissions from electrical equipment are projected to decrease from 327.36 in 2029 to 98.02 kt $CO_2e$ in 2050, making it the most significant end-use sector for emission reductions.

In contrast, Jiangdong Street and Xiaolingwei Street should prioritize emission reduction efforts in residential and public buildings, respectively. In Jiangdong Street, residential buildings are critical targets for emission reduction. By reducing service demand and improving energy utilization, these buildings could achieve a reduction of 429.34 kt $CO_2e$ within the study period. Specifically, in the cooking sector of residential buildings, reducing the use of coal stoves from 8.77% in 2019 to 4.92% by 2050, alongside decreasing reliance on gas and liquefied petroleum gas stoves in favor of electric stoves, could result in a 4.17% reduction in $CO_2$ emissions, lowering them to 6.24 kt $CO_2e$ by 2050. For Xiaolingwei Street, where public land use is predominant, emission reduction should focus on public buildings. Enhancing the energy efficiency of terminal energy-consuming equipment in large public institutions such as universities, schools, research institutes, and associated facilities is crucial. Within the research scope, these efforts could contribute to a reduction of 1010.44 kt $CO_2e$ in the overall emissions from public buildings.

## Discussion

This study develops a city-scale, building-based emissions accounting approach that incorporates building type, function, floor area, and height for individual structures. Additionally, it integrates point source data from power plant facility-level generators to link demand- and supply-side dynamics. This methodology facilitates high-resolution spatial mapping of $CO_2$ emissions during the operational phase of buildings. While the methodological framework maintains global applicability, Nanjing was strategically selected as the case study to demonstrate practical implementation, given its socioeconomic representativeness as a transitional urban economy and pioneering status in low-carbon policy experimentation[44,45]. Our findings indicate that in 2019, $CO_2$ emissions during this phase amounted to 19,015.37 kt $CO_2e$. Notably, there is a marked disparity between commercial and residential buildings: the overall carbon intensity of commercial buildings during the operational phase is 3.9 times higher than that of residential buildings, primarily due to the different types of end-use energy equipment involved.

Under both regulatory and blueprint scenarios, considerable cumulative emission reductions can be achieved through rigorous management of energy use intensity for critical end-use equipment on the demand side, combined with accelerated phase-out of high-energy-consumption, high-emission coal-fired power plants on the supply side. Relative to the regulatory scenario, implementing more stringent energy efficiency standards for commercial HVAC and lighting systems while reducing the operational lifespan of existing coal plants by five years is projected to yield a reduction of 2232.05 kt $CO_2e$ from 2020 to 2050. As demonstrated in Supplementary Fig. 2, commercial buildings emerge as the pivotal sector for demand-side mitigation, with HVAC system efficiency improvements under the regulatory scenario delivering cumulative emission reductions of 4872.96, 13,369.44, and 14,298.41 kt $CO_2e$ during 2021–2030, 2031–2040, and 2041–2050, respectively. The blueprint scenario, incorporating enhanced demand-side measures such as widespread

adoption of energy-efficient lighting and more ambitious efficiency standards, generates an additional 23,698 kt $CO_2e$ reduction compared to the baseline. Comparative analysis reveals that plant retrofits and efficiency upgrades contribute 68.24% of supply-side emission reductions during 2031–2040 under the regulatory scenario, while early coal plant retirements dominate the 2041–2050 period, accounting for 72% of supply-side reductions. The synergistic integration of demand- and supply-side interventions offers transformative mitigation potential, with actionable policy measures including: development of low-energy lighting technologies coupled with application guidelines, deployment of cost-optimized sensor-based lighting controls; enhancement of appliance energy labeling systems complemented by local government incentives for ultra-efficient devices; and advancement of low-/zero-carbon power generation technologies alongside efficiency retrofits of existing plants. Concurrently, maintaining energy security while strategically retiring subcritical coal-fired capacity remains critical to eliminate high-emission generation infrastructure.

This methodology is both innovative and adaptable, offering the ability to integrate various building attributes such as height, area, and function to accurately calculate $CO_2$ emissions during the operational phase of buildings. Moreover, it is adaptable enough to be applied across different cities, providing valuable insights for emission reduction strategies and aiding policymakers in developing informed roadmaps. A limitation of this work is that, due to the absence of detailed building data in rural areas, the constructed database does not fully encompass all relevant building attributes. However, the data we collected covers 534,000 buildings, representing areas where 81.87% of Nanjing's population resides, suggesting that the building data is both representative and relatively accurate.

## Methods

### CEADs building model

In this study, we developed a comprehensive model (CEADs-building model) for building-based emission accounting and the analysis of emission reduction pathways, which integrates both demand and supply-side factors. Specifically, we fused building data collected from multiple sources and classified building types into three categories: residential buildings, commercial buildings, and public buildings, based on the actual usage of each structure. Our urban building morphology dataset comprises vector data including building footprints, geographic coordinates (latitude/longitude), and height information. Building upon established methodologies from recent literature[46–51], we developed a Python-based computational pipeline to: (1) extract Points of Interest (POI) data through the Amap API, (2) perform rigorous data cleaning and classification to obtain location names, functional categories, and commercial attributes, and (3) systematically categorize POIs into standardized classes (dining, retail, finance, education, healthcare, etc). These georeferenced POI data were then spatially joined with building vectors using coordinate matching algorithms to derive building functional types—a critical parameter for emission modeling. Residential buildings encompass both residential and mixed-use housing; commercial buildings include entities such as businesses, restaurants, retail spaces, hotels, and financial services; and public buildings cover structures with functions related to government, education, healthcare, sports, and community services.

Given that $CO_2$ emissions from building operations account for approximately 22% of the national energy-related carbon dioxide emissions[52], this study aims to further elucidate the impact of various end-use energy activities on energy consumption and $CO_2$ emissions during the operational phase of buildings. To achieve this, we propose a modeling framework that is centered around end-use energy activities, with the goal of identifying practical and feasible pathways for decarbonization within the building sector.

$CO_2$ emissions during the operational phase of buildings encompass both direct (Scope 1) and indirect (Scope 2) sources. Direct emissions arise from the consumption of primary energy sources such as coal, oil, and natural gas, which are used in activities like heating and cooking within buildings. Indirect emissions, on the other hand, are associated with the consumption of electricity for lighting, HVAC systems, electrical equipment, and other energy-dependent activities. In residential buildings, specific sources of energy consumption and their associated $CO_2$ emissions include cooking, household lighting, refrigeration, hot water systems, washing machines, and other appliances. For commercial buildings, $CO_2$ emissions are quantified across various sectors, including catering, commercial HVAC, commercial lighting, commercial electrical equipment, and overall commercial power consumption. In public buildings, the $CO_2$ emissions are calculated for public HVAC systems, public lighting, public electrical equipment, and power usage within public institutions. Furthermore, point source data from power plant facility-level generators point sources are incorporated to link and analyze the interactions between demand and supply-side factors. This comprehensive approach ensures that both direct and indirect emissions are thoroughly accounted for across different building types, providing a detailed understanding of the carbon footprint associated with building operations.

To elucidate the $CO_2$ emissions arising from current and future energy consumption during the operational phase of buildings, and to analyze the evolving patterns and trends in emissions under different reduction technologies, this study establishes multiple predictive scenarios. The specific research framework is presented in Supplementary Fig. 4.

## Methods of calculating $CO_2$ emissions

The CEADs-building model consists of four interconnected modules: (1) Energy Demand Module, (2) Energy Conversion Module, (3) Environmental Impact Module, and (4) Socio-economic Module.

(1) Energy Demand Module. Theoretically, a "bottom-up" model structure requires estimating the unit energy consumption for each year throughout the study period. However, in practice, it is often sufficient to define data for the initial year, the final year, and one or more intermediary years. Data for the remaining years are interpolated. For example, assuming the energy intensity of each end-use device is available for 2019, $Y_1$, $Y_2$, ..., $Y_n$ and 2050, the compound growth rate ($C$) for each period can be calculated as follows:

For the period 2019 to $Y_1$:

$$C = \sqrt[Y_1 - 2019]{EI_{Y_1}} - 1 \tag{1}$$

For the period $Y_n$ to 2050:

$$C = \sqrt[2050 - Y_n]{1 + \frac{EI_{2050} - EI_{Y_n}}{EI_{Y_n}}} - 1 \tag{2}$$

Where, $EI_{Y_n}$ and $EI_{2050}$ represent the energy intensity in $Y_n$ and 2050, respectively.

The final energy consumption is calculated as follows:

$$EC_n = \sum_i \sum_j AL_{n,i,j} \times EI_{n,i,j} \tag{3}$$

Here, $n$ denotes the type of building, namely, residential, commercial, or public buildings. $i$ refers to end-use energy equipment, such as cooking, lighting, refrigeration units, and electrical appliances in residential buildings. $j$ represents the energy type: Scope 1 direct emissions include those generated from the use of coal, natural gas, liquefied petroleum gas, and similar sources, while Scope 2 indirect emissions pertain to emissions associated with purchased electricity. $AL$ represents the level of activity as a parameter for energy demand; for instance, in the transport sector, this might be represented by kilometers per passenger, whereas in this study, the proxy indicator is the actual floor area of each building (building footprint multiplied by floor height). $EI$ refers to the annual energy intensity of each end-use equipment.

In this context, the direct energy consumption from Scope 1 is disaggregated, while the indirect energy consumption from Scope 2 is further elaborated in the "Energy Conversion Module".

$$EC_n = EC_{n,scope1} + EC_{n,scope2} \tag{4}$$

$$EC_{n,scope1} = \sum_i \sum_j AL_{n,i,j} \times EI_{n,i,j} \tag{5}$$

In this context, $EC_{n,scope1}$ denotes the direct energy consumption associated with scope 1, while $EC_{n,scope2}$ signifies the indirect energy consumption related to scope 2.

(2) Energy Conversion module. The energy conversion module consist of a facility-level power plant energy conversion module and a transmission and distribution module. In this study, data from facility-level power plants utilizing various fuel types, including coal, gas, oil, and biomass, are incorporated. The total energy consumption for energy conversion is then calculated as follows:

$$ET_s = \sum_m \sum_t ETP_{m,t} \times \left( \frac{1}{f_{s,m,t}} - 1 \right) \tag{6}$$

$s$ denotes the primary energy type, while m represents equipment attributes. In this study, the power plant data include installed capacity, service life, generation efficiency, dispatch rules, and other relevant factors. $t$ refers to the secondary energy type. $ET$ represents the total energy consumption for energy conversion, with $ETP$ denoting the amount of energy consumed during the conversion process. The energy types considered in this study include coal, oil, natural gas, biomass, solar, wind, and others. $ETP_{m,t}$ corresponds to the energy consumption $EC_{n,scope2}$ as formula (2) discussed. $f$ is the conversion efficiency, expressed as a percentage.

For the transmission and distribution modules, there is the following formula:

$$Efficiency_p = 1 - Losses_p \tag{7}$$

Where, $Losses_p$ denotes the percentage loss rate incurred during the transmission and distribution phases, and $Efficiency_p$ quantifies the conversion efficiency of the energy conversion facility.

The methodology for calculating the requisite fuel transmission and distribution for each process is delineated as follows:

$$Input_p = \frac{Output_p}{Efficiency_p} \tag{8}$$

Where $Input_p$ represents the electricity generated and supplied by energy facilities, and $Output_p$ refers to the net electricity delivered to the demand sector, accounting for efficiency losses incurred during the transmission and distribution processes on the supply-side.

(3) Environmental Impact module. For Scope 1 $CO_2$ emissions resulting from direct energy consumption, the following formula is applied:

$$CE_{scope1} = \sum_i \sum_j AL_{n,i,j} \times EI_{n,i,j} \times EF_{n,i,j} \tag{9}$$

Where $CE_{scope1}$ represents the Scope 1 $CO_2$ emissions, and $EF_{n,i,j}$ refers to the emission factor for energy type $j$ consumed by a specific end-use equipment $i$.

For Scope 2 $CO_2$ emissions, which are generated through indirect energy consumption during the building operation phase, the following formula is used:

$$CE_{scope2} = \sum_s \sum_m \sum_t ETP_{m,t} \times \frac{1}{f_{s,m,t}} \times EF_{s,m,t} \qquad (10)$$

Where $CE_{scope2}$ represents Scope 2 $CO_2$ emissions, and $EF_{s,m,t}$ refers to the emission factor associated with the process in which a specific point-source unit m utilizes primary energy $s$ to produce secondary energy $t$.

(4) Socio-economic module. The floor area is to some extent affected by the gross domestic product (GDP). The higher the regional GDP, the higher the income level of residents, which leads to the improvement of residents' consumption power. Within the affordable range, residents tend to choose a better and larger living environment[53–56]. On the other hand, the upper part of the GDP will also promote the development of commerce, which will lead to an increase in the area of commercial buildings and public buildings[57,58]. In addition, since this study focuses on the $CO_2$ emissions generated during building operation, which is closely related to human behavior, the speed of population growth will affect $CO_2$ emissions to a certain extent[59]. Therefore, this paper selects the core macroeconomic indicators, including GDP and population as the key influencing parameters of the socio-economic module in the future forecast.

### Data source
Based on the availability and updating of comprehensive data, this work takes 2019 as the base year and 2020–2050 as the future forecast period. The GDP data underlying the model comes from the Gridded datasets for population and economy under the Shared Socio Economic Pathways published by Jiang et al.[60]. The population data comes from the future city level population forecast data of China under various socio-economic paths published by Zhang et al.[61]. The building data in this study was constructed by using multi-source data fusion methods to crawl building information from sources such as Amap, Baidu Maps, Beike Network, Anjuke, and 58.com. The information of each individual building includes four levels of geographic location information: province, city, district, and street, which are then converted into precise geographic coordinates through forward geocoding and spatial matching methods. In addition, there is also information about the building's intended use, commercial district, floor area, and number of floors, which facilitates further classification research and calculation.

The scenario setting parameters in this article are combined with historical development trends, government planning documents, and academic research results at home and abroad. The activity level and energy intensity data of different terminal energy consuming devices are sourced from government policy research reports and existing literature research (Supplementary Tables 4, 5). Emission factors and standard coal coefficients follow the IPCC 2019 Refinement Guidelines, incorporating regional adaptations for Jiangsu's grid emission factors (2020–2022 average).

### Methodological transferability
While this study focuses on Nanjing as a representative case, the proposed framework exhibits significant potential for adaptation to other urban contexts. Three core components ensure replicability: (i) The validation protocol utilizing open-source built environment data. Ground-level imagery serves as the fundamental data source for identifying individual building functions, with commonly accessible data obtained through online mapping services offering street view images. Notable examples include OpenStreetMap (https://www. openstreetmap.org) and GABLE[62], which provides comprehensive features encompassing both building footprints and façade characteristics. (ii) The standardized POI-to-building-function classification pipeline (Step1 in Supplementary Fig. 4). The accuracy of this method, which integrates POI data with building attributes(with code availability), has been extensively validated across various research studies[63–66]; (iii) This study develops a modular emission accounting framework that integrates four key components: energy demand modeling, energy conversion processes, environmental impact assessment, and socioeconomic dimension analysis. The framework requires locally sourced inputs, including: (a) building footprint data, (b) point-source power plant facility records, and (c) regional energy balance tables to enable algorithmic replication and cross-regional application.

### Data availability
The data supporting the findings of this study are available in the Supplementary Information, and the data generated in this study are provided in the Cloud Repository (https://cloud.tsinghua.edu.cn/d/ 512895b942ba4408ab46/). Building data supporting the findings of this manuscript are also available from the corresponding author upon request.

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

## Acknowledgements

We thank the Carbon Neutrality and Energy System Transformation programme. This research is funded by the National Key R&D Program of China (2022YFE0208700, 2022YFE0208500, and 2023YFE011300). We acknowledge support from the Research Grants Council of the Hong Kong Special Administrative Region, China (AoE/P-601/23-N).

## Author contributions

Y.X.C. collected data, performed formal analysis, developed methodology, created visualizations, and wrote the manuscript. Z.Y.W. and Q.W. designed the research, developed the methodology, and contributed to writing—review and editing. J.M. contributed to research design and provided resources. J.W.H. and S.P.L. participated in methodology development and validation. L.Z. performed validation and contributed to writing—review and editing. P.P.C., D.L.L. and J.B. provided supplementary data and materials. D.B.G. acquired funding, designed the research, developed methodology, supervised the project, and contributed to writing—review and editing.

## Competing interests

The authors declare no competing interests.
