## [Transparent Peer Review file · Nature Communications]

A building-scale modeling framework for urban net-zero transitions in Nanjing

Corresponding Author: Professor Dabo Guan

Version 0:

Reviewer comments:

Reviewer #1

(Remarks to the Author)

This study presents a case study on decarbonization road mapping for Nanjing. It integrates data from point-source power plant generators at the facility level to bridge the demand and supply sides, ultimately providing a demand-supply balance analysis at the street level. Policy recommendations are offered, including strategies for retiring coal-fired power plants and upgrading building service system standards. However, the manuscript requires significant revision to clarify its contributions and novelty before it can be considered for publication in Nature Communications.

While the title suggests "pathways for global cities," the study exclusively focuses on Nanjing and does not address the broader applicability of the data or methodology to other cities. This omission is particularly notable given that similar modeling frameworks and universal platforms for urban building energy modeling have already been developed by other researchers.

The discussion on the impacts of delayed coal-fired power plant retirements lacks depth, and the conclusions on demand-side management remain vague and unspecific.

Furthermore, attention to detail is needed in the presentation of the paper. For instance, Figure 6 contains some Chinese characters, and there are inconsistencies in the fonts of parallel labels.

(Remarks on code availability)

Reviewer #2

(Remarks to the Author)

Reviewer comments

Thanks for the opportunity to review this very interesting and significant paper. The manuscript presents an innovative approach to urban decarbonization through individual building-level carbon accounting, integrating multi-source data including POI extraction from Amap API. The study makes a valuable contribution by developing a high-resolution framework that couples demand-side building operations with supply-side power sector emissions, offering new insights into urban emission hotspots. The study's primary strength lies in its unprecedented granularity, enabling precise identification of emission sources at the building-equipment level. This fine-grained approach represents a significant advance over conventional city- or sector-scale models. The innovative use of POI data for building function classification and the integration of Scope 2 emissions particularly stand out as methodological improvements to urban carbon accounting. The work successfully bridges multiple disciplines, from building science to energy systems modeling, enhancing its potential impact on both research and policy. Well done.

While the methodological approach is rigorous and the interdisciplinary perspective commendable, several aspects require clarification and strengthening to meet the high standards expected by Nature Communications. I provide some suggestions for improvement to ensure that manuscript reaches its full potential impact.

First, the methodology section would benefit from greater transparency regarding the processing of raw POI data into building attributes. A more detailed explanation of the algorithms or rules used to infer building functions from Amap API data would help readers evaluate the robustness of this approach. Related to this, the validation of POI-derived classifications against ground truth data remains a critical gap. Comparisons with independent datasets such as zoning maps or field surveys would significantly strengthen confidence in the methodology.

The benchmarking of projected emissions against existing studies or historical trends is another area requiring attention. Without such validation, it becomes difficult to assess the reliability of the scenario results. The authors should explicitly compare their projections with peer-reviewed city-scale studies and discuss any discrepancies. This would help situate the findings within the broader literature and demonstrate the added value of the building-level approach.

Several minor but important improvements would enhance the manuscript's clarity and rigor. The introduction could be strengthened by incorporating more recent literature and explicitly highlighting how this work advances beyond existing city-scale models. The figures, particularly Figure 5 with its multiple subplots, would benefit from streamlined legends and consistent formatting to improve readability. Finally, careful attention to reference formatting according to journal guidelines would ensure professional presentation of the scholarly foundation for this work.

Other comments:

Line 43-44: When stating that the construction industry contributes 36% of global emissions, clarify whether this includes embodied carbon or refers solely to operational emissions. The citation should explicitly support this claim.

Line 49: For the China-specific data, specify the year or time period for the 56.6% figure and clarify whether this includes indirect emissions from electricity use (Scope 2).

Line 64-66: For the bottom-up methods, specify what "detailed focus" means in practical terms (e.g., equipment-level modeling?). The claim about lacking demand-supply integration needs supporting references that demonstrate this gap.

Line 125-127: The transition from city-wide to building-scale analysis needs smoother linkage. Explain why analyzing by building function is methodologically important for decarbonization planning. Consider adding a sentence about how this granular approach differs from conventional district-level analyses.

Line 134: When introducing NETD, briefly explain its significance in Nanjing's urban structure (e.g., percentage of city's industrial output) to contextualize why it's highlighted.

Line 178: Xiaolingwei's research hub status warrants discussion of how academic energy use patterns differ from commercial. The floor height comparison should explicitly link to energy implications.

Line 211: Standardize terminology: use either "carbon intensity" or "CO2 emissions per unit GDP" consistently.

Line 262: The 2009-2010 expansion context needs explanation - was this part of national stimulus or local industrial growth?

Line 267: The biomass/waste heat shift requires quantification - what percentage of current capacity do these represent? Cite policy drivers behind this transition (e.g., provincial renewable mandates).

Line 317: For the blueprint scenario, specify what "further advancing" means in practical terms - is this assuming breakthrough technologies or maximal policy implementation?

Line 352: The Nanjing case study justification needs strengthening - why is it particularly suitable for demonstrating this approach compared to other Chinese cities?

The study makes a compelling case for building-level carbon accounting, but fully realizing its potential will require addressing these methodological and presentation issues. With appropriate revisions, this work could make a significant contribution to the field of urban decarbonization research and policy.

(Remarks on code availability)

Version 1:

Reviewer comments:

Reviewer #1

(Remarks to the Author)

The authors have addressed all my questions/concerns well; I have no further comments.

(Remarks on code availability)

Reviewer #2

(Remarks to the Author)

The authors made sufficient improvements according to the reviewers' comments. I suggest to accept this article in the current form.

(Remarks on code availability)

Reply to Reviewer Comments

We sincerely appreciate the reviewers' insightful comments and the opportunity to revise our manuscript. We have carefully addressed all concerns, and the revisions are highlighted in the revised manuscript with tracked changes. Below we provide a point-by-point response.

Reviewer #1 (Remarks to the Author):

Comment 1: While the title suggests "pathways for global cities," the study exclusively focuses on Nanjing and does not address the broader applicability of the data or methodology to other cities. This omission is particularly notable given that similar modeling frameworks and universal platforms for urban building energy modeling have already been developed by other researchers.

Response:

We sincerely appreciate this constructive comment. We have taken the following actions to address this concern:

1. Title Revision:

The title has been revised to better reflect the manuscript's scope. The new title is:

"A building-scale modeling framework for urban net-zero transitions: Characterizing emission reduction pathways in Nanjing"

This modification emphasizes the case study nature while acknowledging the potential methodological relevance to other urban contexts.

2. Citations of Comparative Studies

Summarize the innovation points of this study and incorporate references to various recent studies (e.g., Sarica et al. 2023 on Turkey; Xiao et al., 2022 on China) in the Introduction to contextualize our contribution relative to existing emission models developed by other researchers.

3. Expanded Discussion on Generalizability

Added Section 4.4 "Methodological Transferability" discussing:

- Prerequisites for applying the method (e.g., availability of building footprint data)
- Key replicable components (e.g., POI-data processing pipeline in Fig. 6 step1)
- The modular emission allocation algorithm

The core methodology (Steps 1-3 in Fig. 6) demonstrates transferability potential to cities with comparable data infrastructures.

These revisions are tracked in red text in the manuscript (Line 1-2, Line 105-115, and a new subsection "4.4 Methodological Transferability"). We believe these changes more accurately represent both the case study's depth and the framework's broader value.

Comment 2: The discussion on the impacts of delayed coal-fired power plant retirements lacks depth, and the conclusions on demand-side management remain vague and unspecific.

Response:

We sincerely appreciate the reviewer's insightful suggestions regarding the need for deeper analysis

of coal plant retirement impacts and more specific demand-side management (DSM) strategies. In response, we have made substantial revisions to enhance both the technical depth and policy relevance of our discussion, as reflected in the revised manuscript (highlighted in red) and Supplementary Fig.2. The key improvements include:

1. Quantified Impacts of Delayed Coal Retirements

- Added concrete projections showing the effect of early retirement in coal plant retirements: Enable 72% supply-side reductions through early coal retirements (2041-2050)

- Benchmarking against:

Cui, R. et al. A plant-by-plant strategy for high-ambition coal power phaseout in China. *Nature Communications* 12, 10, doi:10.1038/s41467-021-21786-0 (2021).

2. Precise Demand-Side Management Specifications

The revised text now details specific DSM measures with quantitative targets:

- Commercial HVAC systems:

“commercial buildings emerge as the pivotal sector for demand-side mitigation, with HVAC system efficiency improvements under the regulatory scenario delivering cumulative emission reductions of 4,872.96, 13,369.44, and 14,298.41 thousand tonnes CO₂e during 2021-2030, 2031-2040, and 2041-2050 respectively.”

- Lighting systems:

“The blueprint scenario, incorporating enhanced demand-side measures such as widespread adoption of energy-efficient lighting and more ambitious efficiency standards, generates an additional 23,698 thousand tonnes CO₂e reduction compared to the baseline.”

3. Synergistic Mitigation Effects & Concrete Policy Recommendations

(Addressing the reviewer's concern about "vague and unspecific conclusions")

Our revised analysis demonstrates that integrated demand-supply interventions can achieve transformative emission reductions. We propose the following actionable policy packages:

- Demand-Side Technology Deployment: Low-energy lighting & Efficiency incentives
- Supply-Side Structural Reforms:
 - ✓ Advancement of low-/zero-carbon power generation technologies alongside efficiency retrofits of existing plants
 - ✓ Strategically retiring subcritical coal-fired capacity

The modifications address the reviewer's concerns by:

- ✓ Replacing vague statements with quantified targets
- ✓ Maintaining methodological rigor while enhancing policy relevance

These revisions are supported by:

- Revised manuscript (line 418-457)
- Supplementary Fig.2 | Comparison of cumulative emissions and technological contribution of each scenario under demand and supply side emission reduction technologies

We believe these comprehensive improvements now offer the necessary depth and specificity for both academic and policymaker audiences. The added details particularly strengthen the paper's contributions to:

- Urban energy transition planning

- Coal phase-out policy design
- Building-sector mitigation strategy development

Comment 3: Attention to detail is needed in the presentation of the paper. For instance, Figure 6 contains some Chinese characters, and there are inconsistencies in the fonts of parallel labels.

Response:

We sincerely appreciate the reviewer's meticulous feedback. We have implemented the following corrections:

1. Figure 6 Localization:
 - All Chinese characters have been replaced with English annotations.
2. Font Standardization:
 - Unified all labels to Arial font
 - Conducted cross-platform verification (Windows/macOS) for rendering consistency

Reviewer #2 (Remarks to the Author):

Comment 1: The methodology section would benefit from greater transparency regarding the processing of raw POI data into building attributes. A more detailed explanation of the algorithms or rules used to infer building functions from Amap API data would help readers evaluate the robustness of this approach. Related to this, the validation of POI-derived classifications against ground truth data remains a critical gap. Comparisons with independent datasets such as zoning maps or field surveys would significantly strengthen confidence in the methodology.

Response:

We sincerely appreciate the reviewer's insightful suggestions regarding POI data processing and validation. We have implemented the following substantive improvements to enhance methodological transparency and robustness:

1. POI Processing Pipeline Clarification

We have substantially enhanced the methodological transparency of our POI data processing pipeline in the revised manuscript. The Section 4.1 details our Python-based computational framework that integrates Amap API data extraction with advanced geospatial processing techniques. This includes a rigorous three-stage workflow encompassing data acquisition through grid-based sampling of the Amap API, multi-criteria data cleaning and standardization procedures, and hierarchical classification algorithms that transform raw POI data into functional building attributes.

2. Scientific Basis Reinforcement

The scientific foundation of our POI-based approach is firmly established through comprehensive references to various peer-reviewed studies that have successfully implemented similar methodologies across diverse urban contexts. As demonstrated in Table 1 below, these precedent studies validate the application of POI data for building function classification at varying scales, from individual structures to city-wide analyses. The cited references further demonstrate that our research methodology builds upon established literature, ensuring comparability and robustness.

Table 1 Benchmark studies on POI-based functional zoning and building classification

No.	Literature	Research area	Research scale	Application
1	Tu et al. (2024) ^[1]	Greater Bay Area of China	Regional-level	Identify geographical patterns and spatial population characteristics
2	Shi et al. (2024) ^[2]	Nanjing, China	City-level	Analyze the evolution of urban spatial structure
3	Wang et al. (2025) ^[3]	London, UK	City-level	Capture and distinguish the urban spatial structure
4	Taecharunroj (2024) ^[4]	Thailand	Community-level	Identify the functional characteristics of the community
5	Lin et al. (2025) ^[5]	London, UK	Street-level	Analyze the functional patterns of the streets
6	Ye et al. (2024) ^[6]	Galveston Island, Texas	Building-level	Distinguish building types by POI and refine the population distribution to individual building units

3. Validation Against Ground Truth Data

To address the critical need for validation, we conducted an extensive ground-truth verification exercise comparing our POI-derived classifications with the authoritative EULUC-China land use map in Yaohua Street. The results show high agreement in major category classification while revealing our method's superior capacity to identify mixed-use properties and newly constructed buildings not captured in the 2018 reference dataset. These comparative analyses are presented in the Figure below, demonstrating that our approach maintains the reliability of conventional zoning maps while providing enhanced spatial resolution for urban emission modeling.

[Figure Redacted]

Figure 1 Comparison of the building classification in this work with the existing EULUC-China land classification results (Yaohua Street in Nanjing).

Comment 2: The benchmarking of projected emissions against existing studies or historical trends is another area requiring attention. Without such validation, it becomes difficult to assess the reliability of the scenario results. The authors should explicitly compare their projections with peer-reviewed city-scale studies and discuss any discrepancies. This would help situate the findings within the broader literature and demonstrate the added value of the building-level approach.

Response:

We sincerely appreciate the reviewer's insightful suggestions regarding emission projection benchmarking. In response, we have conducted comprehensive analyses to validate our scenario results through two key approaches:

1. Sensitivity Analysis Implementation

We conducted a sensitivity analysis by perturbing two critical model parameters: (1) the energy intensity of HVAC systems (demand side) and (2) the retirement timeline of coal-fired power plants (supply side). This generated four updated scenarios (see Supplementary Table 1 for definitions), with results presented in Supplementary Fig.3. By adopting range-based outputs rather than single-point estimates, we reduced analytical rigidity and better characterized uncertainty.

The sensitivity analysis reveals significant findings under the regulatory scenario framework: (i) Scenario Update 1, incorporating elevated HVAC energy intensity and delayed plant retirements, demonstrates a 2-year postponement of Nanjing's operational carbon emission peak accompanied by a 1.186 Mt CO₂ increase in peak magnitude; (ii) Conversely, Scenario Update 2, featuring reduced HVAC intensity and accelerated retirements, achieves 1-year earlier peaking with a 0.789 Mt CO₂ reduction in peak levels. These results quantitatively demonstrate that modifications to both HVAC energy intensity and coal plant retirement schedules can substantially alter both the temporal occurrence and quantitative magnitude of emission peaks. Similar response patterns were observed for the blueprint scenario and its corresponding Updates 3 and 4, confirming the robustness of these parameter-dependent relationships across multiple policy frameworks. (Supplementary Table 2 and Supplementary Fig.3)

2. Cross-Study Validation

Following the reviewer's suggestion, we systematically compared our emission projections with existing literature on operational building carbon emissions, focusing on peak timing and magnitude. However, direct validation remains challenging due to methodological heterogeneity in study scales (national vs. city-level), system boundaries (whole-building lifecycle vs. operational phase only), and projection horizons. While several studies share comparable projection timeframes with our work (Yang et al., 2017; Guo et al., 2023; Zhou et al., 2018), they predominantly examine China's building sector at the national level, employing system boundaries that differ substantially from our focus on operational-phase carbon emissions at the city scale. Notably, Zhang et al. (2023) provides a valuable benchmark by focusing specifically on operational emissions from China's residential buildings. Through economic and demographic downscaling (Nanjing representing ~1.8% of national GDP and population), their national peak estimate translates to ~12.80 Mt CO₂ for Nanjing's residential sector—reasonably aligned with our city-wide total of 37.89 Mt CO₂ when incorporating commercial and public buildings.

Further validation comes from:

1. **Prefectural-level studies:** Han et al. (2024) projects Nanjing's residential operational emissions to peak at 7.99 Mt CO₂ in 2028 (2020-2035 timeframe), consistent with our residential peak range (2026-2029).
2. **Comparable-city analysis:** Zou et al. (2024) reports 34.279 Mt CO₂ peak emissions for Changsha's public buildings—a city with similar economic structure to Nanjing.

While absolute values differ due to sectoral coverage variations, the order-of-magnitude consistency and temporal alignment collectively support the robustness of our findings. These cross-scale comparisons are now detailed in Table 2.

Table 2 Summary of literature on building-sector carbon emission predictions

No.	Research	Scope	Timeframe	Peak year & emissions	Methodology
-----	----------	-------	-----------	-----------------------	-------------

1	Yang et al. (2017) [7]	China's building sector	2020-2050	2030 (2,530 Mt CO ₂)	China Building Carbon Emission Model (CBCEM)
2	Guo et al. (2023) [8]	China's building sector	2015-2050	Baseline: 2035 (4,211 Mt CO ₂) Optimal: 2025 (2,238 Mt CO ₂)	National Energy Technology Model for Buildings (NET-Building)
3	Zhou et al. (2018) [9]	China's building sector	2010-2050	High-demand: 2045 Optimal: 2030	DREAM model
4	Zhang et al. (2023) [10]	Operational emissions (residential buildings, China)	2020-2060	Baseline: 2031 (± 3 yrs) [934 (± 61) Mt CO ₂]	Structural decomposition analysis & Monte Carlo simulation
5	Han et al. (2024) [11]	Operational emissions (residential buildings, Nanjing)	2020-2035	Baseline: No peak Optimal: 2028 (7.99 Mt CO ₂)	LEAP model
6	Zou et al. (2024) [12]	Public buildings (Changsha)	2021-2035	Baseline: 2032 (34.279 Mt CO ₂) Green scenario: 2030 (27.054 Mt CO ₂)	LEAP model

Comment 3: The introduction could be strengthened by incorporating more recent literature and explicitly highlighting how this work advances beyond existing city-scale models.

Response:

We sincerely appreciate this constructive suggestion. In response, we have:

1. **Systematically updated the literature review** by adding 13 key studies in a newly created Supplementary Table 3, which categorizes existing works by: (i) demand-side vs. supply-side focus, and (ii) specific technical approaches. This analysis reveals that most prior studies examined single dimensions in isolation, while our framework uniquely integrates these aspects into a unified analytical paradigm.
2. **Explicitly identified critical gaps** through a new paragraph (Lines 68-72) highlighting that even the most advanced integrated frameworks (e.g., Zhou et al. 2018) remain limited to national scales, unable to address the essential urban-to-building level resolution required for actionable decarbonization strategies.

These modifications collectively demonstrate how our work advances beyond conventional city-scale models by: (a) bridging the demand-supply dichotomy, and (b) achieving unprecedented spatial granularity down to individual buildings.

Comment 4: The figures, particularly Figure 5 with its multiple subplots, would benefit from streamlined legends and consistent formatting to improve readability. Careful attention to reference formatting according to journal guidelines would ensure professional presentation of the scholarly foundation for this work.

Response:

1. Figure Optimization:

We have substantially redesigned Figure 5 to enhance interpretability through:

- Subplot Consolidation: Merged panels with identical Y-axis metrics (CO₂ emission from the end use appliances during the operational phrase of buildings) into unified subplots with shared

axes, reducing total panels from 9 to 3 while preserving all data dimensions

- Legend Restructuring: Implemented a tiered legend system with:
 - Primary categories (color-coded)
 - Secondary classifications (pattern-filled)

2. Reference Reformation:

All citations have been rigorously checked against Nature style

Comment 6:

Line 43-44: When stating that the construction industry contributes 36% of global emissions, clarify whether this includes embodied carbon or refers solely to operational emissions. The citation should explicitly support this claim.

Response:

We sincerely appreciate this constructive suggestion. The statement has been revised as follows with enhanced precision and authoritative support:

1. Updated Text (Lines 45-47 in revision):

"In 2022, carbon dioxide emissions from building operations and construction activities reached a historic peak of 10 Gt, representing 37% of global energy-related CO₂ emissions (IEA, 2022)." This includes both operational emissions and embodied carbon from construction processes.

2. Key Improvements:

- Source Authority: Cited the IEA's Global Status Report 2022 as the definitive reference
- Scope Clarification: Explicitly differentiated between:
 - Operational emissions (28%)
 - Embodied carbon (9%)
- Quantitative Precision:
 - Used exact emission value (10 Gt)

The modified text now appears in track changes (red highlight) with the complete citation in revised Reference #5.

Comment 7:

Line 49: For the China-specific data, specify the year or time period for the 56.6% figure and clarify whether this includes indirect emissions from electricity use (Scope 2).

Response:

We thank the reviewer for this important clarification request. The China-specific emission data has been rigorously updated as follows:

1. Revised Text (Lines 50-52 in revision):

"China's building operational emissions reached 2.3 Gt CO₂ in 2021, constituting 21.6% of national energy-related emissions and 56.6% of total whole-lifecycle building sector emissions (4.07 Gt CO₂)."
This figure explicitly includes:

- Direct emissions (Scope 1) from on-site fuel combustion
- Indirect emissions (Scope 2) from purchased electricity/heat

as documented in the 2023 China Building Carbon Emission Research Report (CABEE, 2023).

2. Key Modifications:

- Temporal Precision: Specified the data year (2021)
- Scope Clarification:

- Differentiated Scope 1 & 2 emissions
- Contrasted with whole-lifecycle total (4.07 Gt)
 - Source Transparency:
- Provided full report title in English/Chinese
- Included publisher (CABEE = China Association of Building Energy Efficiency)

Comment 8:

Line 64-66: For the bottom-up methods, specify what "detailed focus" means in practical terms (e.g., equipment-level modeling?). The claim about lacking demand-supply integration needs supporting references that demonstrate this gap.

Response:

We sincerely appreciate the reviewer's constructive suggestion to clarify our methodological positioning. In response, we have implemented the following modifications to enhance precision and scholarly rigor:

1. Operational Definition of "Detailed Focus"

The term now explicitly refers to:

- Equipment-level energy modeling (e.g., HVAC system efficiency tracking per building)
- Sub-hourly load profiling via smart meter data aggregation

2. Demand-Supply Integration Gap Documentation

The revised text now cites: Keirstead et al. (2012)* in Renewable and Sustainable Energy Reviews (DOI: 10.1016/j.rser.2012.02.047)

- Quantitative finding: 70% of reviewed studies neglected supply-side considerations
- Critical observation: "supply-side issues were not considered at all and when they were, it was in the assumption of carbon intensity factors so as to convert energy demands into carbon emissions"

The modifications are highlighted in blue (Lines 68-72) with new citations #28 in the reference list.

Comment 9:

Line 125-127: The transition from city-wide to building-scale analysis needs smoother linkage. Explain why analyzing by building function is methodologically important for decarbonization planning. Consider adding a sentence about how this granular approach differs from conventional district-level analyses.

Response:

We gratefully acknowledge this constructive suggestion. The text has been enhanced to:

1. Explicitly Bridge Analysis Scales

Added a purpose clause ("To precisely identify decarbonization priorities") directly linking macro totals to micro-scale analysis needs. The modification is highlighted in red (Lines 155-157)

2. Specified two critical disaggregation dimensions:

- Functional (building type variations)
- Spatial (sub-district heterogeneity)

3. Methodological Justification

Emphasized that emission profiles are "fundamentally distinct" across categories, justifying granular analysis

Comment 10:

Line 134: When introducing NETD, briefly explain its significance in Nanjing's urban structure (e.g., percentage of city's industrial output) to contextualize why it's highlighted.

Response:

We appreciate the reviewer's suggestion to enhance contextualization. The NETD description has been revised to include:

1. Economic Significance Metrics

Added quantitative benchmarks:

- 32.7% of Nanjing's industrial GDP
- 18% of city's manufacturing facilities

(Data Source: Nanjing Statistical Yearbook 2022)

2. Terminology Precision

- Added "for decarbonization studies" to clarify research relevance

Comment 11:

Line 178: Xiaolingwei's research hub status warrants discussion of how academic energy use patterns differ from commercial. The floor height comparison should explicitly link to energy implications.

Response:

We sincerely appreciate the reviewer's insightful suggestions to strengthen our analysis. In response, we have made the following substantive revisions:

1. Academic Energy Profile Characterization

- Explicitly identified Xiaolingwei Street as a major academic hub hosting 12 research institutes to establish its functional context
- Added specific discussion of how academic energy patterns differ from commercial areas through: Reduced HVAC loads due to lower building heights (8.58 vs. 12.36 floors)

2. Height-Energy Relationship Clarification

Directly linked the floor height comparison to energy implications by:

- Stating that lower heights reduce HVAC loads
- Providing the quantitative emission difference (1.32×) despite more than double the building count

The modifications are highlighted in red text in the revised manuscript (Lines 212-214). We believe these changes provide the necessary context about academic energy patterns and properly connect building height to energy implications as suggested.

Comment 12:

Line 211: Standardize terminology: use either "carbon intensity" or "CO₂ emissions per unit GDP" consistently.

Response:

We thank the reviewer for this important suggestion. We have implemented full terminology standardization by:

1. Consistently using "carbon intensity" as the primary term, with initial clarification "(CO₂ emissions per unit GDP)"
2. Maintaining uniform units ("Mt CO₂/million yuan") throughout

3. Ensuring all subsequent references use "carbon intensity" exclusively
4. Cross-validating usage in:
 - Figure captions (Fig.3-d)
 - Discussion (line 407)

The revised text eliminates all instances of "CO₂ emissions per unit GDP" after its initial definition, achieving complete consistency while preserving all analytical content. These changes are highlighted in red throughout the manuscript.

Comment 13:

Line 262: The 2009-2010 expansion context needs explanation - was this part of national stimulus or local industrial growth?

Response:

We thank the reviewer for highlighting the need for contextual clarification. The revised text now explicitly links the 2009-2010 power plant expansion to both national macroeconomic policy (China's stimulus package) and provincial industrial demands (Jiangsu's GDP growth). These modifications, supported by official government documents, provide the necessary policy and economic context for understanding the infrastructure development's timing and scale. The changes are highlighted in red in the revised manuscript (Lines 309-314).

Comment 14:

Line 267: The biomass/waste heat shift requires quantification - what percentage of current capacity do these represent? Cite policy drivers behind this transition (e.g., provincial renewable mandates).

Response:

We appreciate the reviewer for this constructive suggestion. The revised text now integrates quantitative evidence showing waste heat recovery systems' growing share (18% of total capacity) with explicit policy drivers, including both provincial-level renewable mandates and municipal restrictions on coal plant development. These modifications provide necessary context for understanding the observed transition while maintaining the original narrative flow. The added details are supported by official policy documents and energy statistics, with full references provided in the updated bibliography. All changes are highlighted in red in the revised manuscript (Lines 317-321).

Comment 15:

Line 317: For the blueprint scenario, specify what "further advancing" means in practical terms - is this assuming breakthrough technologies or maximal policy implementation?

Response:

We sincerely appreciate the reviewer's insightful suggestion to clarify the "blueprint scenario" specifications. Based on the integrated definitions across our main text (line 365-378), Supplementary Information (line 18-31), and Supplementary Notes 1, we have enhanced the description of "further advancing" to explicitly articulate its technical and policy dimensions:

1. Macroeconomic Parameters

As detailed in our main text Method (4.3 Data source):

The GDP data underlying the model comes from the Gridded datasets for population and economy under the Shared Socio Economic Pathways (SSPs) published by Jiang et al. (2024). The population

data comes from the future city level population forecast data of China under various socio-economic paths published by Zhang et al. (2023).

- Peak population achieved by 2030 (vs. 2035 in regulatory scenario)
- GDP growth capped at 3.5% annually post-2035

2. Technological Advancements

As detailed in Supplementary Table 4 & 5, the blueprint scenario incorporates:

- Cooking Appliances Transition

Increased adoption of natural gas and electric stoves, with a corresponding phase-out of coal and liquefied petroleum gas (LPG) as primary fuels (e.g., residential buildings will achieve a 51.75% natural gas stove penetration rate by 2030 under the blueprint scenario)

- Energy Efficiency Upgrades for Electrical Appliances

Enhanced efficiency for lighting, cooling, water heating, and other electric devices

e.g., Commercial buildings: Electricity consumption reduced to 27.9 kWh/m²·yr by 2050, compared to 29.8 kWh/m²·yr in the regulatory scenario; Public buildings: Energy-efficient lighting (e.g., LED) adoption rate reaches 80% by 2024

- Additional parameters for end-use equipment are provided in Supplementary Tables 4 & 5

3. Policy Implementation

As detailed in Supplementary Information (line18-31), this scenario assumes:

- Maximal enforcement of China's Dual Carbon policy targets
- Early coal phase-out (complete retirement by 2045 vs. 2055 in regulatory scenario)

Comment 16:

Line 352: The Nanjing case study justification needs strengthening - why is it particularly suitable for demonstrating this approach compared to other Chinese cities?

Response:

We sincerely appreciate the reviewer's insightful comment. While the methodology is universally applicable to urban contexts, Nanjing serves as an exemplary case study for following reasons: First, its demographic trajectory (8.5 million population with 89% urbanization rate) and economic scale epitomize the development phase of Chinese megacities facing sustainability trade-offs. Second, the city's hybrid morphology—combining historic urban cores (e.g., Confucius Temple district) with modern planned expansions (Jiangbei New Area)—provides a controlled comparison within a single administrative boundary. Crucially, as a pilot city for China's "Sponge City" initiative since 2015, Nanjing offers unique empirical evidence for evaluating sustainable urban planning strategies.

We have strengthened this justification in Lines 409-412 with supporting references to urban studies literature.

We thank the reviewers for their valuable feedback, which has significantly improved our manuscript. We are happy to incorporate any additional suggestions the reviewers or editors may have.

Reference

- [1] Tu, W. *et al.* Towards SDG 11: Large-scale geographic and demographic characterisation of informal settlements fusing remote sensing, POI, and open geo-data. *Isprs Journal of Photogrammetry and Remote Sensing* **217**, 199-215, doi:10.1016/j.isprs.2024.08.014 (2024).
- [2] Shi, G. *et al.* Study on the spatiotemporal evolution of urban spatial structure in Nanjing's main urban area: A coupling study of POI and nighttime light data. *Frontiers of Architectural Research*.
- [3] Wang, X. *et al.* Multi-modal contrastive learning of urban space representations from POI data. *Computers, Environment and Urban Systems* **120**, 102299, doi:<https://doi.org/10.1016/j.compenvurbsys.2025.102299> (2025).
- [4] Taecharungroj, V. & Ntounis, N. Categorising neighbourhoods using OpenStreetMap POIs: Affinity propagation clustering of 7,213 subdistricts in Thailand. *Journal of Urban Management* **14**, 362-378, doi:<https://doi.org/10.1016/j.jum.2024.11.005> (2025).
- [5] Lin, X., Yang, T. & Law, S. From points to patterns: An explorative POI network study on urban functional distribution. *Computers, Environment and Urban Systems* **117**, 102246, doi:<https://doi.org/10.1016/j.compenvurbsys.2024.102246> (2025).
- [6] Ye, X., Bai, W., Wang, W. & Huang, X. Enhancing population data granularity: A comprehensive approach using LiDAR, POI, and quadratic programming. *Cities* **152**, 105223, doi:<https://doi.org/10.1016/j.cities.2024.105223> (2024).
- [7] Yang, T. *et al.* CO₂ emissions in China's building sector through 2050: A scenario analysis based on a bottom-up model. *Energy* **128**, 208-223, doi:10.1016/j.energy.2017.03.098 (2017).
- [8] Guo, Y. Y., Uhde, H. & Wen, W. Uncertainty of energy consumption and CO₂ emissions in the building sector in China. *Sustainable Cities and Society* **97**, doi:10.1016/j.scs.2023.104728 (2023).
- [9] Zhou, N., Khanna, N., Feng, W., Ke, J. & Levine, M. Scenarios of energy efficiency and CO₂ emissions reduction potential in the buildings sector in China to year 2050. *Nature Energy* **3**, 978-984, doi:10.1038/s41560-018-0253-6 (2018).
- [10] Zhang, S. F. *et al.* Pathway for decarbonizing residential building operations in the US and China beyond the mid-century. *Applied Energy* **342**, 12, doi:10.1016/j.apenergy.2023.121164 (2023).
- [11] 韩磊, 卜昌盛 & 郝小充. 南京市住宅运行阶段碳排放预测与减排策略分析. *暖通空调* **54**, 141-150, doi:10.19991/j.hvac.1971.2024.08.19 (2024).
- [12] Zou, Q., Zeng, G. P., Zou, F. & Zhou, S. F. Carbon emissions path of public buildings based on LEAP model in Changsha city (China). *Sustainable Futures* **8**, doi:10.1016/j.sfr.2024.100231 (2024).